# Polyhexamethylene Guanidine Phosphate Induces Apoptosis through Endoplasmic Reticulum Stress in Lung Epithelial Cells

**DOI:** 10.3390/ijms22031215

**Published:** 2021-01-26

**Authors:** Mi Ho Jeong, Mi Seon Jeon, Ga Eun Kim, Ha Ryong Kim

**Affiliations:** 1School of Pharmacy, Sungkyunkwan University, Suwon 16419, Korea; algh8906@naver.com (M.H.J.); amy951019@naver.com (M.S.J.); 2College of Pharmacy, Daegu Catholic University, Gyeongsan 38430, Korea; kkamy10@naver.com

**Keywords:** PHMG-p, ER stress, apoptosis, lung fibrosis

## Abstract

Airway epithelial cell death contributes to the pathogenesis of lung fibrosis. Polyhexamethylene guanidine phosphate (PHMG-p), commonly used as a disinfectant, has been shown to be strongly associated with lung fibrosis in epidemiological and toxicological studies. However, the molecular mechanism underlying PHMG-p-induced epithelial cell death is currently unclear. We synthesized a PHMG-p–fluorescein isothiocyanate (FITC) conjugate and assessed its uptake into lung epithelial A549 cells. To examine intracellular localization, the cells were treated with PHMG-p–FITC; then, the cytoplasmic organelles were counterstained and observed with confocal microscopy. Additionally, the organelle-specific cell death pathway was investigated in cells treated with PHMG-p. PHMG-p–FITC co-localized with the endoplasmic reticulum (ER), and PHMG-p induced ER stress in A549 cells and mice. The ER stress inhibitor tauroursodeoxycholic acid (TUDCA) was used as a pre-treatment to verify the role of ER stress in PHMG-p-induced cytotoxicity. The cells treated with PHMG-p showed apoptosis, which was inhibited by TUDCA. Our results indicate that PHMG-p is rapidly located in the ER and causes ER-stress-mediated apoptosis, which is an initial step in PHMG-p-induced lung fibrosis.

## 1. Introduction

Lung fibrosis is a chronic interstitial disease characterized by a progressive decline in lung function and premature death from respiratory failure. At the mechanistic level, lung fibrosis is considered to be the result of a dysregulated wound-repair response to lung injury. Repeated microscopic injuries cause the death of epithelial cells, which is followed by an inflammatory process, and then the accumulation and persistence of activated reparative mesenchymal cells (myofibroblasts) that are responsible for the deposition of excessive extracellular matrix [1,2,3]. Evidence supports that the injury of epithelial cells is an initial event of fibrogenesis [4,5]. Moreover, the importance of epithelial cells in the pathogenesis of lung fibrosis is implicated by the observation that inhibition of epithelial cell death attenuates the extent of lung fibrosis in animal models [6,7,8].

The endoplasmic reticulum (ER) is a cellular organelle responsible for the biosynthesis, folding quality control, maturation, and trafficking of proteins. Any condition that perturbs protein processing can lead to the accumulation of misfolded proteins in the ER, in a condition termed ER stress [9]. In response to ER stress, the signaling cascade called the unfolded protein response (UPR) orchestrates adaptive cellular changes to reestablish homeostasis. If stress resolution fails, the UPR commits the cell to apoptotic death [10]. Prolonged or excessive ER stress is associated with the development and progression of lung fibrosis through induction of apoptosis in epithelial cells. Korfei et al. [2] demonstrated that in human lung tissues, the apoptotic pathway was activated in the areas of fibrosis that prominently expressed markers of ER stress. Moreover, animal models have shown that ER stress contributes to fibrotic remodeling in the lungs [11,12,13]. 

Polyhexamethylene guanidine phosphate (PHMG-p), a cationic disinfectant, has been shown to be strongly associated with lung fibrosis in epidemiological and toxicological studies [14,15]. Our previous studies showed that PHMG-p induced inflammatory and fibrotic responses, but also p53-mediated cell cycle arrest and apoptosis [16,17,18]. However, the molecular mechanism in epithelial cell death induced by PHMG-p is currently unclear. We synthesized a PHMG-p–fluorescein isothiocyanate (FITC) conjugate and assessed the uptake into lung epithelial A549 cells. To examine intracellular localization, the cells were treated with PHMG-p–FITC, counter-stained with various cytoplasmic organelles, and observed by confocal microscope. Moreover, the organelle-specific cell death pathway was investigated in cells treated with PHMG-p.

## 2. Results

### 2.1. PHMG-p Is Rapidly Located in Endoplasmic Reticulum of Lung Epithelial Cells

Airway epithelial cells are the cells lining that primary entry portal of inhaled agents into the body. Past studies on the mechanism of action of guanidine cationic polymers have focused on their ability to interact with membranes. However, Chindera et al. [19] demonstrated that polyhexamethylene biguanide (PHMB), another guanidine polymer, can enter different kinds of mammalian cells. In a detailed study, we identified the subcellular localization of PHMG-p in lung epithelial cells. A549 cells with stained ER, lysosomes, or mitochondria were incubated with 2 μg/mL PHMG-p–FITC for different periods of time. There was no significant cytotoxicity at any of the time points (data not shown).

In Figure 1, PHMG-p–FITC and each organelle are shown in green and red, respectively. Areas stained yellow were indicative of PHMG-p–FITC localization in subcellular components. Confocal images of the cells in Figure 1a show that PHMG-p–FITC was localized in the ER after incubation for more than 1 h. After incubation for 5 h, PHMG-p–FITC was also localized in the lysosomes (Figure 1b), but not in the mitochondria (Figure 1c). The fluorescence emissions of PHMG-p–FITC coincide with that of the ER-tracker dye, indicating that PHMG-p–FITC is rapidly localized in the ER, not in other subcellular components. 

### 2.2. PHMG-p Induced ER Stress after Internalization into ER

To demonstrate the induction of ER stress by PHMG-p, we evaluated the expression of ER stress components over time (Figure 2 and Figure A1). First, the expression levels of BiP and phosphorylated PKR-like ER kinase (PERK) among UPR canonical pathways increased dose-dependently after 1 h of PHMG-p exposure (Figure 2a). After 3 h of PHMG-p exposure, p-eIF2α and ATF4, which were dependent on phosphorylated PERK, were upregulated (Figure 2b). Moreover, CHOP increased as an ER stress-derived pro-apoptotic factor after 6 h of exposure (Figure 2c). In other ER stress-associated pathways including IRE1 and ATF6, they showed non-significant responses after PHMG-p exposure (Figure A1). In vivo, the expression of CHOP was increased both 4 and 7 days after intratracheal instillation of PHMG-p (Figure 2d).

### 2.3. PHMG-p Activated Mitochondria-Associated Apoptotic Pathway

Since PUMA is a well-known p53 modulator induced by ER stress, we identified an increase in PUMA expression after 6 h of PHMG-p exposure (Figure 3a). PUMA can regulate apoptosis via the modulation of mitochondrial outer membrane permeabilization (MOMP) [20]. The effect of PHMG-p on mitochondrial membrane potential was quantified using JC-1 staining. At high membrane potentials, JC-1 forms J-aggregates showing red fluorescence instead of green fluorescence. The decrease in the intensity ratio of red/green fluorescence was observed in A549 cells exposed to PHMG-p (Figure 3b). After 24 h of PHMG-p exposure, p-p53 and Bax were increased, while Bcl-2 levels decreased (Figure 3c). Furthermore, the expression of p-p53 and cleaved caspase 3 was upregulated by PHMG-p administration in vivo (Figure 3d). Moreover, in the slide sections of mouse lungs, we found that the signals referring to p-p53 (green) and CHOP (red) were increased and co-localized in the context of the intratracheal instillation of PHMG-p (Figure 3e).

### 2.4. ER Stress Inhibitor Attenuates ER Stress and Apoptosis Induced by PHMG-p

To validate that PHMG-p-induced ER stress triggered apoptosis, we pretreated A549 cells with tauroursodeoxycholic acid (TUDCA) as an ER stress inhibitor, and then exposed them to PHMG-p. When pretreated with TUDCA, ER stress components, including p-eIF2α, ATF4, and CHOP, decreased in the presence of PHMG-p (Figure 4a). Moreover, PUMA was attenuated. After 24 h of PHMG-p exposure, p-p53 was also decreased by TUDCA (Figure 4b), which implied that decreased ER stress alleviated the apoptotic pathway of PHMG-p. Flow cytometry results demonstrated that TUDCA decreased the percentiles of both early apoptotic and late apoptotic cells from 15.02% and 15.90% to 6.76% and 10.10%, respectively (Figure 5). In the same manner as the result of TUDCA pretreatment, the PERK inhibitors (GSK2606414 and GSK2656157) attenuated PHMG-p-induced apoptosis (Figure A2). 

## 3. Discussion

Several toxicological studies have demonstrated that inhaled PHMG-p induces epithelial injury, inflammation, and fibrosis in the respiratory tract. However, limited information is available about the mechanism of epithelial injury induced by PHMG-p. The present study suggested a possible mode of action of inhaled PHMG-p in the respiratory tract. PHMG-p and FITC conjugates indicated that PHMG-p was rapidly located in the ER of lung epithelial cells. The regulation of signaling pathways associated with ER stress and apoptosis was identified in A549 cells and mice exposed to PHMG-p. In addition, ER stress-mediated apoptosis was verified using TUDCA. 

The literature contains conflicting evidence and interpretations regarding the interaction between guanidine polymers and membranes. Previous studies have shown that cationic guanidine groups interact with negatively charged phospholipids in the bacterial membrane [21]. This electrostatic interaction causes a leakage of low molecular weight cytoplasmic components and activation of membrane-bound enzymes, followed by an extensive disruption of the cytoplasmic membrane. As a study consistent with this, Choi et al. [22], using the fungus *Candida albicans*, reported that polyhexamethylene guanidine hydrochloride (PHMG-h), a similar guanidine polymer with a different salt from PHMG-p, exerted antifungal activity with a membrane-targeted pore formation mechanism. On the other hand, two individual experimental studies using PHMB appear to be consistent in reporting the non-disruptive translocation of PHMB across the membrane by several processes such as endocytosis [19,23]. Endocytosed materials are transported to the early endosomes, where they can be sorted for recycling back to the plasma membrane or targeted downstream to the late endosomes, followed by maturation to or fusion with lysosomes [24]. In the present study, PHMG-p–FITC, as applied to A549 cells, was rapidly located in the ER, followed by the lysosomes (Figure 1). We speculated that, like PHMG-h, PHMG-p–FITC might form stable or transient pores in zwitterionic phosphatidylcholine and partially penetrate into A549 cells. Since phosphatidylcholine is the major glycerophospholipid (~60%) in the ER [25], it also accounts for the internalization of PHMG-p–FITC into the ER. Additionally, PHMG-p–FITC was certainly translocated by endocytosis, partially recycled by the early endosomes, and partially found in the lysosomes after 5 h. Further studies are needed to fully elucidate the behavior of PHMG-p in the airway epithelium.

Figure 2 shows that PHMG-p internalized in the ER places a burden on the function of the ER by activating the PERK signal-transduction branch of UPR. Upon ER stress, PERK, one of three ER transmembrane proteins, is dimerized and autophosphorylated to become active. PERK activation then directly leads to the phosphorylation of eIF2α, which is associated with the suppression of most mRNA translation by blocking the binding of initiator Met-tRNA to the ribosome. However, ATF4 mRNA is selectively translated under these conditions, thereby upregulating the transcription of downstream genes, including the apoptotic factor CHOP [26]. In most reports, CHOP is primarily induced downstream of PERK, but all three branches of the UPR can participate in CHOP induction [27]. After all, the PERK branch of UPR is strongly protective at moderate levels of ER stress, but can contribute to cell death pathways, mediated by its downstream effector, CHOP. Several studies have suggested that there is some uniqueness in the PERK-governed signaling pathway in inducing apoptosis compared to signaling pathways by other ER transmembrane proteins such as IRE1 and ATF6. Liu et al. [28] found that, although all three branches of ER stress were activated, PERK deletion exhibited a stronger protective effect against apoptosis than either IRE1 or ATF6. In addition, PERK has been considered a component of the mitochondria-associated ER membrane [29]. This area is described as the physical and functional contact site of the ER and mitochondria, which are tightly packed with proteins with various functions, including cell death regulation [30]. Therefore, it is possible that the PERK-governed signaling pathway activated by PHMG-p carries out more responsibilities in conducting death signals. 

Importantly, our data indicated that PHMG-p activated the mitochondria-associated apoptotic pathway (Figure 3). This is consistent with our previous report showing that silencing p53 attenuated PHMG-p-induced apoptotic markers, including caspase-3 and PARP [16]. Additionally, PUMA induction was identified in the present study. Although the CHOP-related apoptotic signaling network is complex, CHOP is known to induce PUMA by binding to the *PUMA* promoter during ER stress [31]. PUMA primarily activates Bax or Bak, which in turn induces a selective process of MOMP through the formation of channels or pores after oligomerization [32]. This process was indirectly demonstrated via JC-1 staining, indicating that damage to the mitochondrial membrane occurred in response to PHMG-p (Figure 3b). Based on our results and scientific findings, we assumed that PHMG-p-induced ER stress was implicated in the apoptosis of A549 cells. Notably, TUDCA, an ER stress inhibitor, attenuated the eIF2α/ATF4/CHOP signaling pathway, and affected the expression of PUMA in response to PHMG-p (Figure 4a), indicating that TUDCA inhibited PHMG-p-induced ER stress. Moreover, it resulted in a decrease in p-p53 levels (Figure 4b) associated with the inhibitory effect on PHMG-p-induced apoptosis (Figure 5 and Figure A3). In addition, PHMG-p-induced early apoptosis was dose-dependently attenuated by two kinds of PERK inhibitors (Figure A2), which demonstrated the apoptotic effect of induced PERK activation by PHMG-p. Taken together, our results suggest that PHMG-p caused ER stress-mediated apoptosis in A549 cells. 

Multiple studies have confirmed that ER stress is involved in the pathogenesis of lung fibrosis by increasing epithelial cell apoptosis. Representative lung fibrogenic agents, including bleomycin and paraquat, induce the apoptosis of lung epithelial cells through the activation of the PERK-governed signaling pathway [33,34]. This study was consistent with these reports on fibrogenic agents. These findings indicated that PHMG-p was rapidly located in the ER and caused ER-stress-mediated apoptosis, which was implied as an initial step in PHMG-p-induced lung fibrosis. In addition, TUDCA was able to affect the ER stress pathway from PHMG-p-induced apoptosis, providing an alternative strategy for the future treatment of chemical-induced lung fibrosis.

## 4. Materials and Methods 

### 4.1. Cell Culture

Human lung epithelial cells (A549) were obtained from the Korean Cell Line Bank (Seoul, Korea) and cultured in RPMI 1640 supplemented with 5% fetal bovine serum (Biotechnics Research Inc., Lake Forest, CA, USA), penicillin (100 units/mL), and streptomycin (100 mg/mL) at 37 °C in an atmosphere of 5% CO_2_/95% air under saturating humidity.

### 4.2. Synthesis of PHMG-p–FITC Conjugates

The conjugate PHMG-p–FITC was synthesized according to the method of Chindera et al. [19]. PHMG-p was provided by the Korea Institute of Toxicology (Jeongeup, Korea). About 2 mg of fluorescein isothiocyanate (FITC; Sigma-Aldrich, St. Louis, Mo, USA) was dissolved in 800 μL dimethyl formamide (Sigma-Aldrich) containing 50 μL of N,N-diisopropylethylamine (Sigma-Aldrich). The mixture was combined with 200 μL aqueous PHMG-p and shaken overnight at room temperature. The resulting solution was dialyzed using a molecular weight cut off membrane of 3.5 kDa against 50% aqueous ethanol for 5 days with an intermittent change of the dialysate (10 times, 500 mL), and lyophilized to obtain fluoresceinyl–PHMG-p (PHMG-p–FITC). 

### 4.3. Confocal Laser Scanning Microscopy of A549 Cells

A549 cells were seeded onto cover slips in 24-well plates at a density of 10 × 10^4^ cells/well. After incubation for 24 h at 37 °C, the cells were treated with PHMG-p–FITC (2 μg/mL) for time periods. Following incubation, cytoplasmic organelles such as the ER, lysosomes, and mitochondria were stained using ER-Tracker™ Red (Thermo Fisher Scientific, Waltham, MA, USA), LysoTracker™ Red DND-99 (Thermo Fisher Scientific), and MitoTracker™ Red CMXRos (Thermo Fisher Scientific), respectively, according to the manufacturer’s protocol. After organelle staining, 4% paraformaldehyde was added to the cells for 10 min at 37 °C, followed by washing 3 times with phosphate-buffered saline (PBS). Fluorescence emissions were observed using a Zeiss LSM 700 Laser Confocal Microscope (Thornwood, NY, USA). 

### 4.4. Western Blot

A549 cells were seeded in a 6-well plate (15 × 10^4^ cells/well), incubated for 24 h at 37 °C, and then treated with PHMG-p at concentrations of 0–4 μg/mL for different periods of time. The cells were washed twice in PBS and lysed with a radioimmunoprecipitation assay buffer (#89901, Thermo Fisher Scientific) with a protease inhibitor cocktail (GenDEPOT, Barker, TX, USA), phosphate inhibitor (BioVision, Nilpitas, CA, USA), and 0.1% SDS. Lung tissues were homogenized in a tissue protein extraction reagent (#78510, Thermo Fisher Scientific) with a protease inhibitor cocktail on ice. Samples were harvested into a 1.7-mL microcentrifuge tube and incubated on ice for 30 min. After incubation, the mixtures were centrifuged at 14,000× *g* for 15 min at 4 °C. The lysates were quantified using the Micro BCA™ protein assay kit (Prod#23235, Thermo Fisher). The samples were denatured with a buffer containing 2% SDS, 6% 2-mercaptoethanol, 40% glycerol, 0.004% bromophenol blue, and 0.06 M Tris-HCl at 90–100 °C for 6 min, and cooled at room temperature for 5 min. The denatured total proteins (15–25 μg) were loaded onto 10% acrylamide gels. After electrophoresis at 80 V for 20 min and 120 V for a further 60 min, the proteins were transferred onto 0.2 mm polyvinylidene fluoride (PVDF) membranes (170–4156; Bio-Rad Laboratories) using the Trans-Blot^®^ Turbo system (BIO-RAD Laboratories). The membranes were blocked with 5% skim milk in TBS-T for 1 h at room temperature (20 °C) and subsequently incubated with the appropriate primary and secondary antibodies. The membrane was incubated with enhanced chemiluminescence (ECL) reagents (170–5060; BIO-RAD Laboratories) for 5 min and developed using an automatic X-ray film processor (JP-33; JPI Healthcare, Seoul, Korea). The densities of each band were normalized to those of the GAPDH band. The primary antibodies used were rabbit anti-Bip (1:1000 dilution; #3183; Cell Signaling Technology), rabbit anti-phosphorylated PERK (1:1000 dilution; #3179; Cell Signaling Technology), mouse anti-PERK (1:1000 dilution; sc-377400, Santa Cruz Biotechnology, Inc., Santa Cruz, CA, USA), rabbit anti-phosphorylated eIF2α (1:1000 dilution; #3597; Cell Signaling Technology), rabbit anti-ATF4 (1:1000 dilution; #11815; Cell Signaling Technology), mouse anti-CHOP (1:1000 dilution; #2895; Cell Signaling Technology), rabbit anti-PUMA (1:1000 dilution; #4976; Cell Signaling Technology), rabbit anti-phosphorylated p53 (1:1000 dilution; #9284; Cell Signaling Technology), rabbit anti-Bcl-2 (1:1000 dilution; sc-492, Santa Cruz Biotechnology, Inc., CA, USA), rabbit anti-Bax (1:1000 dilution; sc-493, Santa Cruz Biotechnology, Inc., CA, USA), rabbit anti-Caspase 3 (1:1000 dilution; #9665; Cell Signaling Technology), and mouse anti-GAPDH (1:20000 dilution; 015-25473; Wako pure chemical industries, Osaka, Japan). The secondary antibody used was a horseradish peroxidase-conjugated anti-mouse IgG (AAC10P; Serotec, Oxford, UK), diluted 1:10,000.

### 4.5. JC-1 Staining

The mitochondrial membrane potential of the cells was determined using the classical JC-1 (Thermo Fisher Scientific) staining method [35]. Briefly, following PHMG-p treatment, cells were harvested, washed twice with PBS, and then incubated with 500 μL JC-1 staining solution (5 μg/mL) for 20 min at 37 °C in darkness. Next, the cells were suspended with trypsin and analyzed using a flow cytometer.

### 4.6. Flow Cytometry Analysis for Apoptosis

A549 cells were seeded into a 60-mm dish at a density of 60 × 10^4^ and incubated for 24 h at 37 °C. After pretreatment with 50 μM TUDCA for 3 h, A549 cells were exposed to 0–4 μg/mL PHMG-p for 24 h. Following incubation, cells were washed twice with cold PBS and then centrifuged at 1000 rpm at 4 °C for 5 min. The collected cells were suspended in 1 mL of PBS and re-centrifuged at 1000 rpm at 4 °C for 5 min. After discarding the supernatant, the cells were resuspended in a binding buffer (V13242; Thermo Fisher Scientific, Waltham, MA, USA) and 100 μL of the solution containing the cells was transferred into a 1.7-mL microcentrifuge tube. Next, 5 μL of FITC Annexin V (V13242; Thermo Fisher Scientific) and 1 μL PI (V13242; Thermo Fisher Scientific) were added. The solutions were gently mixed and incubated for 15 min at room temperature in the dark. After the incubation period, 400 μL of binding buffer was added to each sample. Keeping the samples on ice, the samples were analyzed by flow cytometry (Guava easyCyte 8HT Flow Cytometer; Luminex, Austin, TX, USA) within 1 h.

### 4.7. Animal Experiment

All animal experiments were approved by the Sungkyunkwan University Institutional Animal Care and Use Committee (Approval No.: IACUC2020-07-04-1, Approval Date: 30 July 2020) and established in accordance with the guidelines of the National Institutes of Health. Male C57BL/6 mice (7 weeks old; 22–24 g) were purchased from Raonbio (Gyeonggi, Korea). Water and food were provided ad libitum. After 1 week of adaptation, the mice were anesthetized with isoflurane, and each mouse was intratracheally administered with 50 μL PHMG-p diluted in saline at a single dose of 1.5 mg/kg. At days 4 and 7, the mice were sacrificed, and lung tissue samples were collected for analysis.

### 4.8. Statistics

Data were analyzed using Excel^®^ 2013 (Microsoft, Redmond, WA, USA) and SigmaPlot^®^ 12.0 software (Systat Software Inc., San Jose, CA, USA). Data from each assay are expressed as mean ± standard deviation. Statistical analysis was performed using SPSS version 21.0 (SPSS, Chicago, IL, USA). Differences between groups were assessed by Duncan’s post-hoc test after one-way analysis of variance. Statistical significance was accepted for values of *p* < 0.01.

## Figures and Tables

**Figure 1 ijms-22-01215-f001:**

Localization of polyhexamethylene biguanide phosphate (PHMG-p)–fluorescein isothiocyanate (FITC) in the subcellular components of lung epithelial A549 cells. A549 cells with stained (**a**) endoplasmic reticulum (ER), (**b**) lysosomes, or (**c**) mitochondria incubated with 2 μg/mL of PHMG-p–FITC for different periods of time. Images show cellular staining of PHMG-p–FITC, organelle staining, a micrograph overlay showing sites of co-localization, and a differential interference contrast micrograph. Areas stained yellow are indicative of PHMG-p–FITC localization in subcellular components.

**Figure 2 ijms-22-01215-f002:**
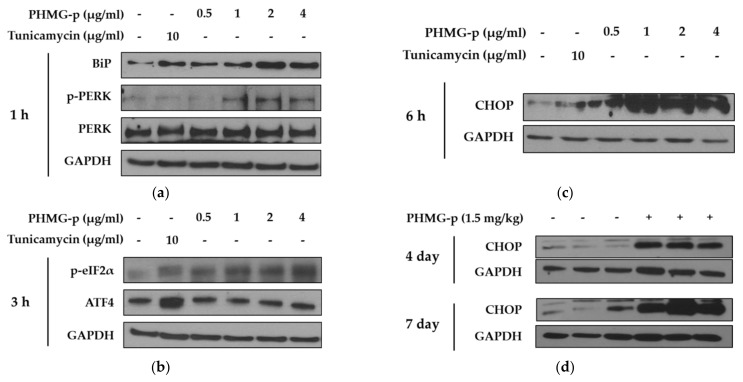
ER stress induced by PHMG-p in A549 cells and mice. To evaluate the effect of PHMG-p exposure on ER stress, the expression of ER stress components was measured depending on exposure time and PHMG-p concentration. We exposed PHMG-p for (**a**) 1, (**b**) 3, and (**c**) 6 h. (**d**) The expression level of CHOP was evaluated with Western blotting after 4 and 7 days of intratracheal administration in mice.

**Figure 3 ijms-22-01215-f003:**
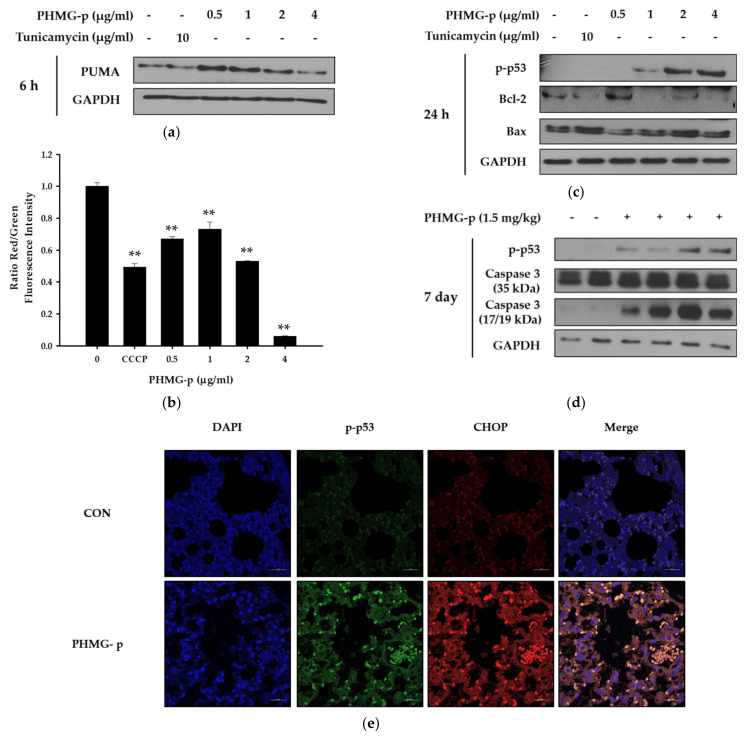
Mitochondria-associated apoptotic factors induced by PHMG-p in A549 cells and mice. (**a**) After 6 h of PHMG-p exposure, expression of PUMA related with ER stress was evaluated with Western blotting. (**b**) JC-1 dye was applied for quantification of the altered mitochondrial potential by exposure to PHMG-p for 24 h. Carbonyl cyanide 3-chlorophenylhydrazone (CCCP) was used as a positive control. (**c**) Expression of apoptosis-associated factors including p-p53, Bcl-2, and Bax were evaluated after 24 h of PHMG-p exposure. (**d**) In mice, the expression of apoptotic factors such as p-p53 and cleaved caspase 3 was measured with Western blotting. (**e**) Seven days after the intratracheal administration of PHMG-p (1.5 mg/kg), lung tissues were subjected to immunohistochemical staining (blue, nucleus; green, p-p53; red, CHOP). Each value represents mean ± standard deviation. ** *p* < 0.01 versus the control.

**Figure 4 ijms-22-01215-f004:**
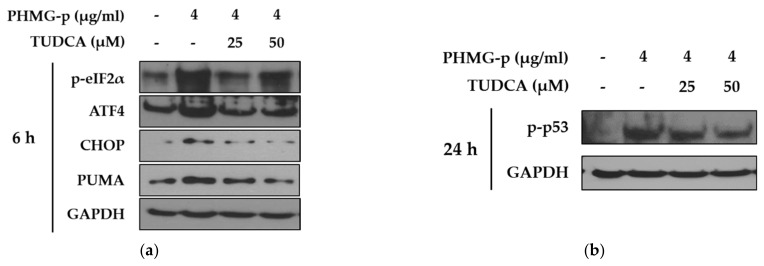
Attenuation of PHMG-p-induced apoptotic pathway by an ER stress inhibitor. By pretreating A549 cells with tauroursodeoxycholic acid (TUDCA), an ER stress inhibitor, we demonstrated its effect on PHMG-p-induced ER stress and apoptosis. (**a**) The expression ER stress components and PUMA was measured after exposure to TUDCA and PHMG-p for 6 h. (**b**) The anti-apoptotic effect of TUDCA was confirmed after 24 h of PHMG-p exposure.

**Figure 5 ijms-22-01215-f005:**
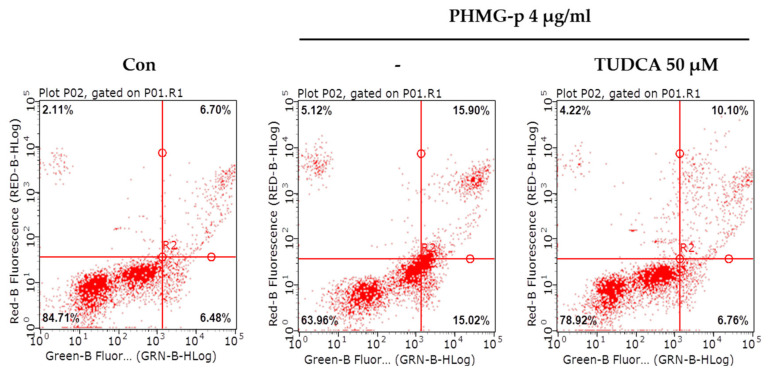
Attenuation of PHMG-p-induced apoptosis by an ER stress inhibitor. After pretreatment with an ER stress inhibitor, TUDCA, we demonstrated its effect on PHMG-p-induced apoptosis. Using flow cytometry, the same number of cells in each group was stained with propidium iodide and annexin V, and analyzed for the percentage of apoptotic cells after exposure to TUDCA and PHMG-p.

## Data Availability

The data presented in this study are available in insert article and supplementary material here and on request from the corresponding author.

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
