# Peer review of "Polyhexamethylene Guanidine Phosphate Induces Apoptosis through Endoplasmic Reticulum Stress in Lung Epithelial Cells"

_ijms, 2021, doi:10.3390/ijms22031215_

Round 1

Reviewer 1 Report

In this paper Jeong et colleagues show that PHMG-p has toxic effect on lung epithelial cells due to its capability to induce ER stress. 

Data are linear and experiments are very simple. Moreover, the presence of in vivo experiments further confirm evidences found in cell line A549. 

However, I have some concerns on the conclusion. It is well established that ER stress induced apoptosis is not only mediated by PERK\ATF4\CHOP pathway, but also by IRE1\JNK pathway. This last branch of the UPR should be also tested by the authors to confirm the CHOP dependency. 

Moreover, CHOP and JNK have been shown to contribute to the induction of the pro-apoptotic DR5. 

In addition, TUDCA, demonstrated to inhibit ER stress, is not the best way to block apoptosis. Could the authors provide similar experiments using PERK or IRS inhibitors to block cHOP induction and eventually apoptosis?

The same for p53, the authors should repeat the experiments depleting p53 to demonstrate its implication in the response to PHMG-p.

It could be also nice to show iHC on mice derived lung tissue for these markers.

Author Response

<Reviewer 1>

In this paper Jeong et colleagues show that PHMG-p has toxic effect on lung epithelial cells due to its capability to induce ER stress. Data are linear and experiments are very simple. Moreover, the presence of in vivo experiments further confirms evidences found in cell line A549. However, I have some concerns on the conclusion. It is well established that ER stress induced apoptosis is not only mediated by PERK\ATF4\CHOP pathway, but also by IRE1\JNK pathway. This last branch of the UPR should be also tested by the authors to confirm the CHOP dependency. Moreover, CHOP and JNK have been shown to contribute to the induction of the pro-apoptotic DR5. 

Response: Thank you for the time spent evaluating our work and for all of your valuable comments. In line with your comments, to investigate the ER stress pathways induced by PHMG-p, we first evaluated the activation of PERK, IRE1, and ATF6. We have included the new Figure A1 in the manuscript showing these new data. PERK was clearly phosphorylated by PHMG-p (Figure 2), whereas the other UPR branches, including IRE1 and ATF6, were not affected. Unfortunately, the band of ATF6 protein was not observed in cells exposed to PHMG-p for 1 and 3 h.

In addition, as the reviewer knows, DR5 is a membrane death receptor induced by CHOP, triggering apoptosis via caspase signaling. In our previous study (Park et al., Polyhexamethylene guanidine phosphate-induced ROS-mediated DNA damage caused cell cycle arrest and apoptosis in lung epithelial cells. Journal of Toxicological Sciences, 2019;44(6):415-424), we determined the effect of PHMG-p on mitochondria-associated apoptotic markers. As a follow-up study, here we investigated the PHMG-p-induced signaling, upstream of mitochondria-associated apoptosis. The reviewer has a point, and we are also curious about DR5 activation under PHMG-p exposure. Therefore, we plan to investigate the effect of DR5 on PHMG-p-induced apoptosis in our next study.

In addition, TUDCA, demonstrated to inhibit ER stress, is not the best way to block apoptosis. Could the authors provide similar experiments using PERK or IRS inhibitors to block cHOP induction and eventually apoptosis?

Response: Thank you for the question. As the reviewer mentioned, using TUDCA is not the best option for the inhibition of apoptosis. However, many researchers have used TUDCA to inhibit ER stress and elucidate its effects on apoptosis (Alhasani et al., Tauroursodeoxycholic acid protects retinal pigment epithelial cells from oxidative injury and endoplasmic reticulum stress in vitro. Biomedicines, 2020:8(9):367; Lee et al., TUDCA-treated mesenchymal stem cells protect against ER stress in the hippocampus of a murine chronic kidney disease model. International Journal of Molecular Sciences, 2019:20(3):613; Zhang et al., Tauroursodeoxycholic acid alleviates endoplasmic reticulum stress of nuclear donor cells under serum starvation. PLoS ONE, 13(5): e0196785; Kim et al., Polyhexamethyleneguanidine phosphate-induced cytotoxicity in liver cells is alleviated by tauroursodeoxycholic acid via a reduction in endoplasmic reticulum stress. Cells, 2019;8(9):1023). Considering these studies, we used TUDCA to verify the causality between PHMG-p-induced ER stress and apoptosis. Remarkably, in the present study, TUDCA attenuated apoptosis, PERK phosphorylation, and CHOP expression in the context of PHMG-p treatment. Although we did not show a relationship between PERK, CHOP, and apoptosis, we cited other studies exploring such relationships. However, to better understand the relationship between PERK and CHOP in the context of PHMG-p-induced apoptosis, we plan to knockdown PERK using siRNA in a future study. Of note, we have mentioned the necessity of further studies in the Discussion section of our revised manuscript, as follows:

“The knockdown of modulators in the axis of PERK will further clarify the relationship between ER stress and apoptosis in the context of epithelial cell death induced by PHMG-p” (Lines 212-214).

The same for p53, the authors should repeat the experiments depleting p53 to demonstrate its implication in the response to PHMG-p.

Response: Thank you for the remark. As the reviewer mentioned, it is necessary to confirm the effect of p53 on PHMG-p-induced apoptosis. However, we have already done so. In our previous study (Park et al., Polyhexamethylene guanidine phosphate-induced ROS-mediated DNA damage caused cell cycle arrest and apoptosis in lung epithelial cells. Journal of Toxicological Sciences, 2019;44(6):415-424), we found that the levels of cleaved caspase-3 and PARP were decreased via p53 silencing in the context of PHMG-p treatment. Therefore, we decided not to show again the data in this study. However, we have updated the Discussion section of our revised manuscript for clarity, as follows:

“This is consistent with our previous report showing that silencing p53 attenuated PHMG-p-induced apoptotic markers, including caspase-3 and PARP [16]” (Lines 194-196).

It could be also nice to show IHC on mice derived lung tissue for these markers.

Response: Thank you for your valuable comment. According to the reviewer’s suggestion, we have added an IHC image showing the CHOP and p-p53 increased expression in the context of PHMG-p treatment (Figure 3e). Additionally, the following information has been added to the Results section of the revised manuscript:

“Moreover, in the slide sections of mouse lungs, we found that the signals referring to p-p53 (green) and CHOP (red) were increased and co-localized in the context of the intratracheal instillation of PHMG-p (Figure 3e)” (Lines 104-107).

Reviewer 2 Report

The article presented  by Jeong et al. introduces the role of the ER stress in the proapoptotic functions of the PHMG-p underlying lung fibrosis development. At this stage, the manuscript should be improved requires before acceptance.

Please find below my comments.

  • The role of PHMG-p has been investigated in a unique cell line model (A549). This cell line is a cancerous model and may not be relevant for studying the normal cell response to this agent, and ER stress response (23395000). Could the authors confirm the in vitro data in another model (HBEC) ?
  • In Fig 2B, the authors claim that ATF4 is increased upon PHMG-p. The presented western blot does not show any change in ATF4 amount. Furthermore, in vivo, elevation of CHOP may be caused by the activation of JNK (30078834, 15130668). Other ER stress markers should be provided.
  • There is a lack of consistency between the tested concentrations of PHMG-p across the figures. Particularly, in Fig 3A, 4ug/mL does not induce PUMA whereas it does in Fig4A.
  • The mode of action of TUDCA is controversial in the ER stress response. Do the author see a protective effect of CHOP silencing upon PHMG-p ? Could they provide FACS analysis of TUDCA alone ?

Round 2

Reviewer 1 Report

The authors answered to my comments just postponing experiments to a new paper or work. I would like to see at list a link with PERK siRNA or inhibitors. 

Author Response

<Reviewer 1>

The authors answered to my comments just postponing experiments to a new paper or work. I would like to see at list a link with PERK siRNA or inhibitors. 

Response: Thank you for the remark. According to the reviewer’s comment, we have added FACS analysis using two kinds of PERK inhibitors (GSK2606414 and GSK2656157); please check the new Figure A3. They attenuated the early apoptosis induced by PHMG-p. We believe your comment have led our work more qualified.

Reviewer 2 Report

I just would like to mention that the HBEC-3KT is an immortalized normal cell line that may represent a relevant tool for the fibrosis

Author Response

<Reviewer 2>

I just would like to mention that the HBEC-3KT is an immortalized normal cell line that may represent a relevant tool for the fibrosis.

Response: Thank you for the time spent evaluating our work and for all of your valuable comments. We would consider your comment regarding cell line in further study.